# Learning-by-Doing Safety and Maintenance Practices: A Pilot Course

**Giovanni Mazzuto** , **Sara Antomarioni**, **Giulio Marcucci** *, **Filippo Emanuele Ciarapica** and **Maurizio Bevilacqua**

Department of Industrial Engineering and Mathematical Science, Università Politecnica delle Marche, 60131 Ancona, Italy
* Correspondence: g.marcucci@staff.univpm.it

**Abstract:** This paper presents an educational approach for teaching Industry 4.0 concepts to maintenance and safety operators involved in industrial processes. A Learning-by-doing approach was introduced to assess the impact of learning by doing and knowledge sharing on designing maintenance and safety solutions based on Industry 4.0 concepts to build experience and improve decision-making skills. To this end, we proposed a pilot course to train industrial operators in the field of new technologies so that they could continue their work effectively. Specifically, the development of the course began with a needs assessment of the perspective participants, followed by an outline of the objectives and course structure. The course was adapted to the different educational and technical backgrounds of the participants (i.e., experienced operators who were digital immigrants and non-experienced operators who were digital natives). The results of the course were assessed through a survey, which allowed us to evaluate the operators' perception of the learning approach and the contribution to improving the operators' competencies and abilities. The results highlighted that the educational approach facilitated the teaching of maintenance and safety principles, promoting operators' attention and participation. The difference in the learning level that we observed between the younger and older operators was also highlighted by the survey results. A dichotomy was revealed between the younger operators, who showed a greater understanding of the explained technologies, and the older operators, who required longer learning times. In this way, both types of participant could benefit from mutual collaboration and teamwork to improve their respective weaknesses.

**Keywords:** Industry 4.0; digital twin; learning by doing; maintenance; safety; learning factory

## 1. Introduction

The industrial and scientific communities are increasingly focused on the challenges associated with Industry 4.0, especially with regard to future trends in the development of industry towards smarter production processes. These include cyber–physical (CPS) and cyber–physical manufacturing (CPPS) systems, which are designed to successfully achieve the intelligent factory paradigm. The adoption of such technologies is now perceived as the key to industrial success, given the possibility of interconnecting machines and tools to improve performance and efficiency. However, the mere introduction of such innovations is not enough; the shared support of the whole company is necessary to achieve success [1]. In this context, if innovation can be considered an opportunity for companies, ad hoc training is the real challenge. There is a need to formulate training courses that are oriented as much as possible towards the practical understanding of these new technologies, taking into account the predisposition of both students who have little practical knowledge but are more accustomed to theoretical study and technological innovations and technicians who are more experienced but have less theoretical knowledge at their disposal. Indeed, according to Čižiūnienė and Batarlienė [2], technicians are one of the most important sources of added value for a company. All activities require the contribution of an operator,

and the more the operator is able to respond to the company's needs, the more easily the activity will be integrated into the company processes. Hence, the process of training an operator becomes a fundamental part of company operations. Moreover, it is essential to emphasize how the learning process is bidirectional: while proper training in industrial practice improves operator performance, the sharing of different cognitive backgrounds, original thoughts, and unusual suggestions can be an additional benefit for the company. Thus, innovative teaching methods to provide company managers with more practical experience are greatly needed [3]. Simulation games for engineering management and project-based education [4] may be considered to maintain a balance between theory and practice.

According to Kafai [5], learning by doing can be used as an anchor for learning by design to reinforce learners' creativity. This approach can involve players in forming, experimenting with, interpreting, and adapting a playing strategy to solve problems, thus enabling players to practice persistent problem solving [6]. Klopfer et al. [7] noted that learning by doing can take the form of dynamic systems, through which players can observe and play out the key principles inherent in the systems and hence develop organizational and systemic thinking skills. The constructivist problem-based and inquiry learning methods have demonstrated the success of learning in the context of challenging, open-ended problems [8]. In any given educational situation, the learning task needs to be presented in a way that is engaging and meaningful to the student and that promotes positive expectations for the achievement of learning objectives [9]. Indeed, training is objectively the meeting point between research and technological and organizational systems. Under these conditions, the objective of education is to offer the best possible service in the field of knowledge transfer, in harmony with technological development [10].

In this context, this work aims to present an educational approach for teaching two of the most important topics of the last ten years, as argued by Agnusdei et al. [11]: Industry 4.0 and digital twins (DTs). Specifically, according to [12], "*the term Industry 4.0 was used to cover two different meanings: as a synonym for an alleged fourth industrial revolution — following those triggered by steam-powered mechanization, electricity and information, and communication technologies (ICT)—and also as a label for the strategic plan pursued by Germany to strengthen its international competitive position in manufacturing*". According to [13], on the other hand, DT is "*a virtual representation of a physical system (and its associated environment and processes) that is updated through the exchange of information between the physical and virtual systems*". Thus, DT makes it possible to apply predictive plant management and maintenance policies. Moreover, DT is reactive, so it changes when the asset changes, and vice versa. For this reason, the implementation of DT is a promising opportunity to increase safety, as process simulation can support operators' training and improve industrial resilience. Company operators must apply these concepts in a laboratory plant that has been developed specifically for the process plant's maintenance and safety.

The proposed learning approach was tailored to industrial engineering operators involved in maintenance and safety processes, due to the fundamental importance of this area in the Industry 4.0 environment [14] and the fact that this was one of the most important aims of the described project. One of our goals was to enhance operators' design capabilities and competencies in industrial systems engineering, stressing the implementation of digital twin and Industry 4.0 principles for improving the reliability and safety of process plants. Another goal of this work was to analyze the learning differences between operators with more than ten years of seniority (expert) and those with less than ten years of seniority (inexpert). Younger operators, in general, have higher technological knowledge baselines, and it is expected that they will also have a higher speed of learning [15]. Thus, the main novelty introduced by the proposed research can be summarized as follows:

(i)　Implementing a training course for industrial operators in the field of new technologies so that they can continue their work effectively;

(ii)　Adapting the course to the different educational and technical backgrounds of the participants.

Hence, the development of a training course adapted to their different aptitudes will enrich from a theoretical point of view those with more practical experience and will benefit from a practical point of view the profiles of the recipients who are less technically proficient but more grounded in theory, allowing mutual exchange and collaboration between the participants. Indeed, judging by the valuable contributions that have already been made to the literature, there is a lack of analysis in terms of courses devoted to different categories of participants and to their collaboration on a theoretical and practical level, and there is a need to extend such courses to topics such as the use of digital twins and safety management.

The following sections are dedicated to explaining the proposed educational approach. Specifically, Section 2 is devoted to a brief literature review of the existing approaches and the identification of the research gaps, and Section 3 outlines the methodology. Section 4 highlights the structure of the plant and outlines the main aims of the project, while Section 5 is dedicated to a detailed course description. The results are summarized in Section 6, while conclusive remarks are drawn in Section 7.

## 2. Literature Review

### 2.1. Learning Strategies in the Current Technical Landscape

According to the existing literature, several different teaching approaches can be found at all levels of education. Several authors found that the conventional lecture approach is followed most of the time, leading to the low involvement of students [16]. People usually process the information they receive in different ways, so attention and focus can easily be lost, and participation in class activities can be missed [17,18]. To this end, alternative learning approaches have been proposed by researchers in this field, with active learning being one of the most popular methods. Active learning is based on constructivism, a learning theory whereby students are engaged in learning the subject matter and pushed to be proactive knowledge creators, abandoning the passive behavior typical of the conventional learning processes [19,20]. The learning-by-doing method, as argued in [21], should be considered in order to improve the quality of learning and surpass traditional learning approaches. In this way, students are stimulated to solve situational problems and develop critical thinking, emulating the professional situations that they will experience in the working environment [22].

A further noteworthy consideration is the addressee of the courses. Prensky [23] differentiates between digital natives (i.e., people born after 1980 who grew up using technologies such as computers, smartphones, and videogames daily) and digital immigrants (i.e., people born before 1980, prior to the digital revolution, who have adopted such technologies for work purposes) [24]. Indeed, different attitudes and behaviors can be observed between different generations when dealing with technologies and the digital environment. These differences must be considered when organizing courses, taking account of the predisposition of the two groups to new technologies, their background knowledge, and their general training in the proposed topic [25]. Indeed, if critically analyzing and solving a problem is a fundamental skill [26], mastering the basic concepts and theories is also needed to establish a baseline: this is the critical added value conferred by the training course proposed in this study.

### 2.2. Learning Factories for Students' and Operators' Training

In recent years, learning factories have been implemented in both the academic and industrial fields as successful manifestations of the active learning paradigm [27] wherein students and practitioners can develop skills. Considering the existing contributions, learning-by-doing approaches—in general—and learning factories—in particular—can be considered emerging research areas, which aim to integrate theory and practice [4], stimulate problem-solving skills [5], and conduct self-assessment evaluations before and after the experience [4,5]. Some authors have noted that involving students in practical activities not only enhances their comprehension of a topic but also improves their critical thinking [6,7]. Several contributions to the literature have addressed this idea. Specifically,

in [6], qualitative comments on the survey outcomes were collected, and the authors were able to assess the approval of the proposed business simulation. In [28], an integrated learning-by-doing approach was proposed to improve upon traditional software teaching for university students. Likewise, in [29], students could learn to adapt a production system to respond to market changes by, for instance, implementing a reconfigurable manufacturing assembly line. The complete process of product development, from customer order to the manufacturing of the product, through to production process adjustment, was incorporated in [30] to offer broader industrial experience to students. In [31], the authors proposed a holistic model for teaching and practicing the integration between shop-floor and top-level decision making. The training was aimed at students working in SMEs so that they could contribute to improving the current production processes and adopting modern technologies. A hybrid plant simulating the production of liquid soap was proposed in [32], aimed at improving the skills of both students and practitioners in the development of new technologies. The implementation of a pneumatic cylinder in a gear motor factory was the basis of the learning factory proposed in [33]: the authors aimed to integrate an existing learning plant by introducing Industry 4.0 tools and techniques, which served as a basis for the transfer of the digital vision to company practitioners.

Similarly, in the smart factory proposed in [34], several technologies typical of the Industry 4.0 paradigm were introduced, e.g., SCADA and PLC programming, IoT, digital twin, smart production, and management systems. In this way, students could be trained to implement such technologies in the industrial environment. At the same time, practitioners could learn which technology best fits their needs and how to implement it. In [35], the authors proposed a framework to guide companies using a learning factory experience. They started with an assessment tool to understand the company's maturity level in order to identify the best learning roadmap and adapt to the specific case, and, finally, they achieved practical learning by providing experience through the learning factory. The authors also proposed a self-evaluation assessment that allowed the participants to scale their confidence levels regarding the different Industry 4.0 tools presented during the workshops, though they did not present the results of the tools' actual application. Among the Industry 4.0-related technologies, virtual reality (VR) has been proposed as a means to enhance training in manufacturing assembly [36], as it appears to speed up the learning process, help learners avoid mistakes during the process, and reduce the gap between the real and digital environments [37]. Assembly and disassembly skills can also be taught in this way, as was proposed in [38], wherein college students participated in a user test to verify the characteristics of the proposed tool and expressed their satisfaction with its usability after the activity through a Likert-scale-based questionnaire. A similar study was also carried out in an industrial environment [39]: the authors presented the results of a usability test involving 26 users divided into three groups, with an ANOVA test applied to the responses.

Most existing studies provide only a description of the learning factories or learning-by-doing approach adopted, thus leaving a research gap. Moreover, only a few works have presented the results achieved through the implementation of the proposed learning approaches (e.g., through self-evaluation questionnaires [3,4,37] or qualitative interviews [5]). This paper will present a description of the facility for which the learning course is proposed and dedicate space to disclosing the participants' learning journey, taking into account their backgrounds and their perceptions of the usefulness of the lessons received. Finally, the results of the self-evaluation questionnaires completed by the participants before and after the course—following the approach proposed in [4]—and the evaluations provided by the plant's external experts who observed the training will be assessed to determine the effectiveness of the pilot course.

## 3. Methodology

This paper describes a project aimed at defining a reference model based on digital twin methodologies for risk reduction in process plants.

In order to standardize the training course properly and disseminate and utilize the project results, the following steps were proposed, as shown in Figure 1: (i) needs identification for training; (ii) training objectives identification, (iii) training course design, (iv) training course delivery, and (v) training course evaluation.

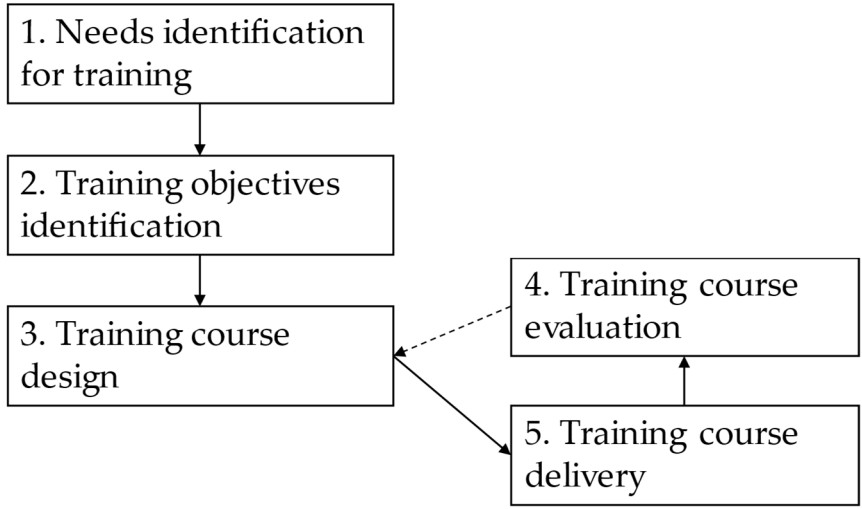

**Figure 1.** Methodology.

The first and fundamental step is "needs identification for training": this lays the foundations and indicates the direction for project outputs by creating a knowledge baseline. To achieve this, it is necessary to evaluate operators' skills, knowledge, and experience while identifying possible useful hints for improvement and thus perfecting the objectives of the training [40]. This information can be obtained by observing practitioners carrying out a specific activity or applying their know-how and determining if and where there is a skill or knowledge gap that should be addressed. In this way, the potential areas of improvement aligned with companies' objectives are also identified [41].

Subsequently, the output of the first phase is used as a complementary input for the second step, "training objectives identification". Indeed, the initial training programs should horizontally provide the knowledge needed for staff to work in specific roles and positions, i.e., basic introductory training on topics such as safety hazards and risks along with their control measures, safe working practices, actions to take in response to safety events, and job-specific training [42]. Moreover, project-related tasks should be highlighted and extensively practiced so that they are performed correctly and consistently. Finally, practical activities should be conducted, e.g., the use of smart sensors and digital twin technology, in order to provide the trainee with a complete overview of the training subject through problem-based learning [43].

In the third step, the course structure is designed, with the aim of ensuring the effectiveness of the work and improving skills, behavior, and performance. In order to design the appropriate training and development scheme, it is necessary to determine what kind of training is required and how it will benefit the organization; therefore, it is necessary to define the training design in detail. During this phase, collaboration with companies, e.g., business-based workshops or company visits, is implemented in order to add a complementary approach to the course topics [44]. Moreover, since dissemination is one of the objectives and crucial activities of this research project, the involvement of end-users and stakeholders in a continuous improvement methodology steers the work towards successful application. Moreover, such actors can also be further involved as members of an advisory board or user group in charge of evaluating the results and providing feedback [45].

Subsequently, the course structure is delivered, i.e., the fourth step: this is the most direct chance for instructors to aid students in acquiring new skills. Effective communication, the incorporation of active learning techniques, and receptivity to learner feedback may boost the impact of the training on the knowledge and abilities of the participants [46].

The training course object of this research paper is exemplified in Section 6.

The final step, performed at the end of the course, is training course evaluation. At this point, all course participants are asked to provide feedback on the training. The survey's structure is based on that proposed in [4]. Specifically, the issues to be addressed include:

- Level of achievement of course objectives;
- Relevance of the topics addressed;
- Level of satisfaction with the theoretical lessons and trainer;
- Level of satisfaction with practical lessons;
- Level of satisfaction regarding the facilities and laboratories.

The participants' feedback on how the training met their needs and the quality of the course can be used to improve the design and delivery of the training [47].

## 4. Case Study

This paper describes a project aimed at defining a reference model based on digital twin methodologies for risk reduction in process plants. The reference model aims to provide a company with all the tools necessary to create a parallelism between the virtual and physical processes to guarantee the static and dynamic analysis of the process. Moreover, it allows the creation of an interconnected system of digital objects, increasing the security of the actors involved and sharing the information generated. Finally, it ensures the possibility of predicting plant faults in good time for a prompt intervention, thus minimizing all possible damage from breakage [48].

### 4.1. The Experimental Plant

The plant in which the project was implemented is an experimental plant located in the Department of Industrial Engineering and Mathematical Sciences (DIISM) of the Polytechnic University of Marche (Ancona, Italy). It reproduces a classic extraction process system. A "new" well, whose pressure is higher than the transport pressure, is used to create suction in a well with insufficient pressure for in-line transport. Indeed, transporting two-phase gas–liquid mixtures in a single pipeline is critical in the oil industry. Hence, using an ejector, a well with a pressure higher than the transport pressure is used to continually extract from a nearby well, which would have to be decommissioned without expensive pumps. Figure 2 shows several components of the plant that was the object of the course participants' practical activities.

### 4.2. Experimental Plant Retrofitting and Consequent Criticalities

An incorrect operation action, such as opening a valve on a system that is supposed to be empty or welding on a container that still holds flammable vapors, can lead to a severe risk of injury or even accidents that can affect the whole company. Thus, the digitalization of the proposed experimental plan could be useful for risk reduction.

If the plant in question is old, a retrofitting process could be the optimal solution to adapt it to Industry 4.0. For this reason, the experimental plant was retrofitted according to the scheme proposed in Figure 3.

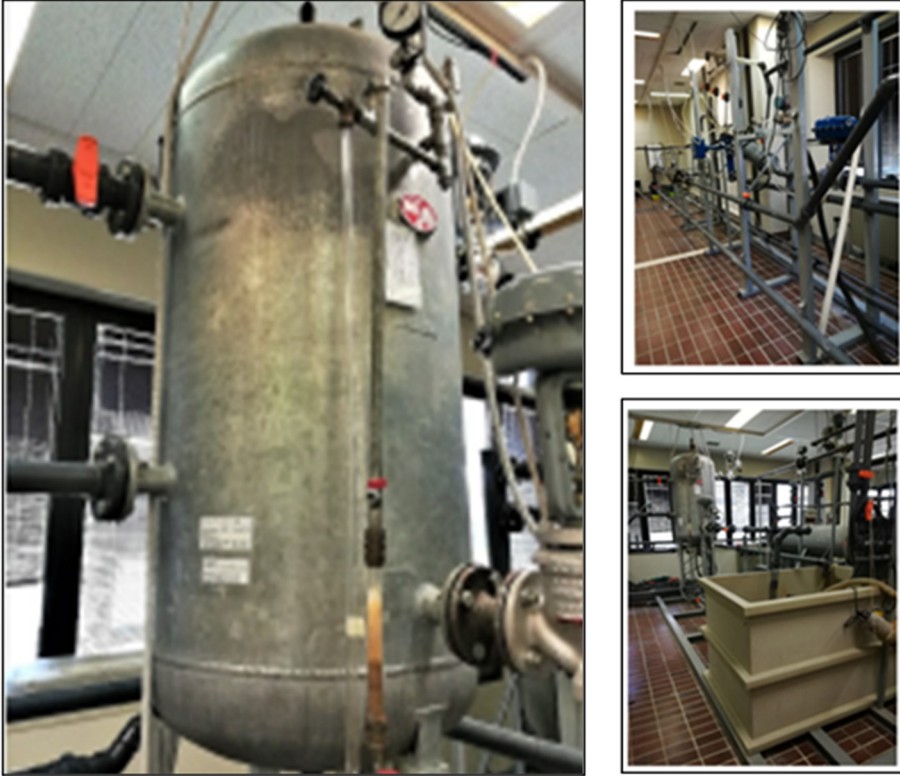

**Figure 2.** Pictures of the experimental plant where students performed the practical activities.

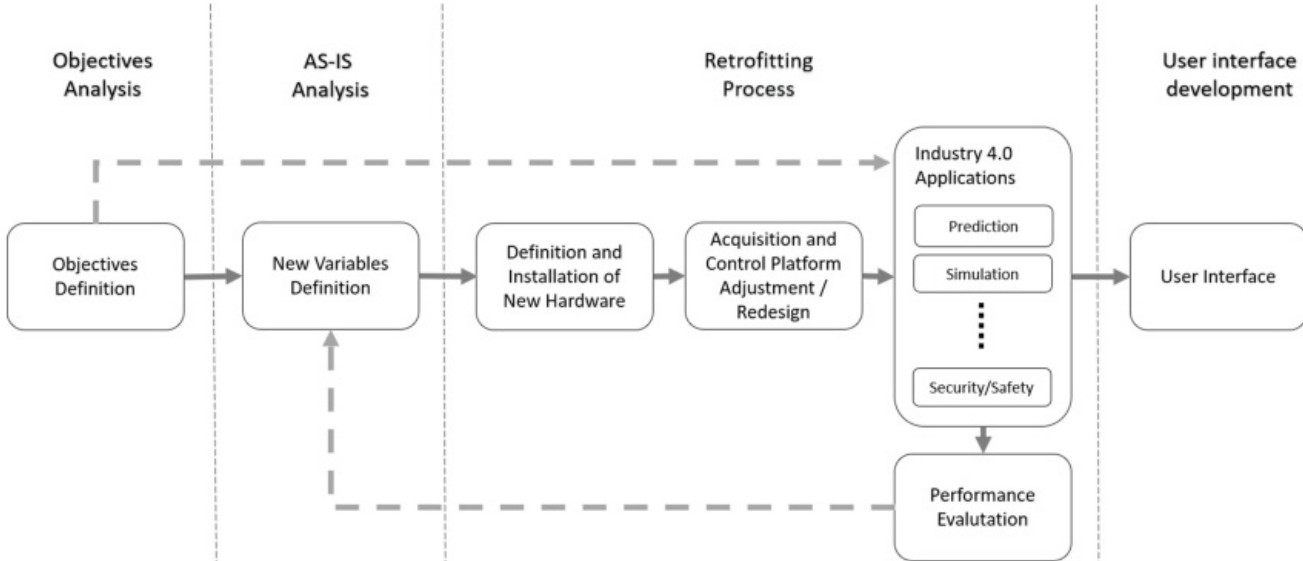

**Figure 3.** Retrofitting of the experimental plant discussed by Di Carlo et al. [49].

Di Carlo et al. [49] explained that applying the pillars of Industry 4.0 in a traditional context confers many benefits in multiple areas, such as efficiency and safety. However, the transition requires the adoption of highly digitized, interconnected, and technologically advanced machinery with unsustainable costs, especially for small and medium-sized enterprises. Moreover, it is necessary to define the system requirements, including the improvement of the working conditions and the implementation of an interconnected communication system. Then, a detailed current plant status description allows the identification of the main process functions and the variables that are monitored. Finally, the new hardware that has been identified can be implemented, and a platform can be developed to

acquire data, control the plant, and simulate new scenarios. Finally, a user interface can be developed to make the developed applications easy to use for less experienced operators.

However, these measures are only effective if correctly applied. Although the digital technologies implemented in the system are intended to reduce the risks faced by operators, they become risk factors if operators are not adequately trained in their use.

It is necessary to consider the abovementioned distinction between operators according to their relationship with digitalization in order to properly train both "digital natives" and "digital immigrants", paying attention to their different approaches to understanding and accepting technology. Specifically, digital natives have access to networked digital technologies and the skills to use them. Most of their daily activities take place and are enhanced with the help of digital technologies. Conversely, digital immigrants were introduced to and adopted technology later in life. It is also true that many digital immigrants have transformed themselves into expert users of digital technology, but, at the same time, their attitudes toward technology differ from those of digital natives [50,51].

### *4.3. The Framework Implementation*

In this section, the framework's main components are described to explain the objectives of each lesson in the training course.

#### 4.3.1. Ejector Digital Twin and Fault Detection System

Despite the widespread use of ejectors in the process industry, predicting the motor and aspirated fluid characteristics, especially in multiphase mixtures, to meet the process requirements and avoid performance problems is complicated.

For this reason, the realization of an ejector model could enable operators to overcome these difficulties by allowing the real-time control of system performance. These arguments prompted Mazzuto et al. [52] to create the digital twin of an ejector for multiphase flows based on swarm intelligence algorithms.

The realized DT allows the creation of virtual process models. It can work online or offline, i.e., inputs can come from sensors (in the first case) or be entered manually (in the second case). In the online mode, the DT receives information from the sensors on the physical asset and changes when the asset is modified. In the offline mode, through a virtual representation of the physical assets, it is possible to carry out scenario analysis without the need to physically implement the process, thus avoiding potential risk situations for operators.

Thus, as described in [53], a test campaign was conducted under normal and abnormal operating conditions to develop the fault detection platform. Manual valves, arranged within the plant, were used to simulate faults, occlusions, and leaks.

The problem of anomaly identification was divided into two sub-problems:

- Assessing whether the system is in a fault condition;
- If the system is in a fault condition, classifying the type of fault.

The first problem was tackled using a traditional approach. Once the data were standardized and the control limit calculated using the steady-state data, it was possible to discriminate the out-of-control points in the tests and assign them to the specific anomaly.

#### 4.3.2. Augmented Reality for Maintenance Instrument

Building on the pillars of Industry 4.0, industrial environments seek to adopt increasingly digitized solutions to achieve so-called smart production. The concept of the augmented operator is part of this approach. The augmented operator can interact with the system with the help of digital tools that facilitate his/her daily work. For this reason, a security and safety application was implemented using AR smart glasses, which were also tested on the experimental plant, as described in [54]. Depending on the physiological characteristics of the operator, the smart glasses can be worn directly on prescription glasses, or the APP can be used on a different android device.

The Vuzix Blade smart glasses (Figure 4) used a cloud architecture connected to the plant. They acted as a guidance system for the operator wearing them, providing remote support that alerted the operator in real time to any hazardous situations. The aim was to assist the operator during the work process, especially concerning maintenance activities.

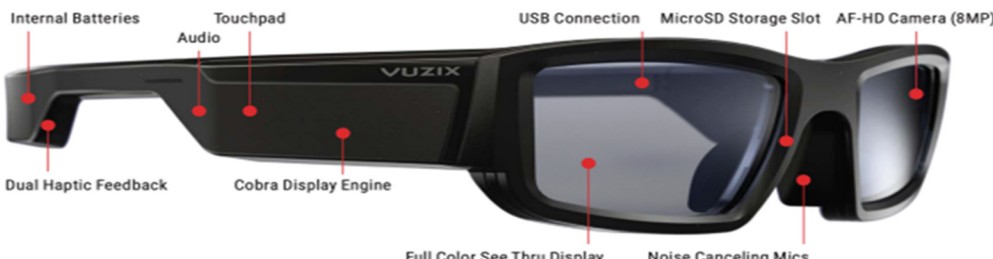

**Figure 4.** Vuzix Blade smart device used in the project implementation.

To summarize, this technology has three main functions:

1.  **Training**: markers are placed at specific points in the plant to advise workers on what to do (Figure 5a);
2.  **Alerting**: the technology, thanks to its cloud-based architecture, displays notifications in cases of an emergency (Figure 5b);
3.  **Remote operator**: the user communicates visually and directly with the control station, receiving real-time assistance and guidance in the correct execution of the task.

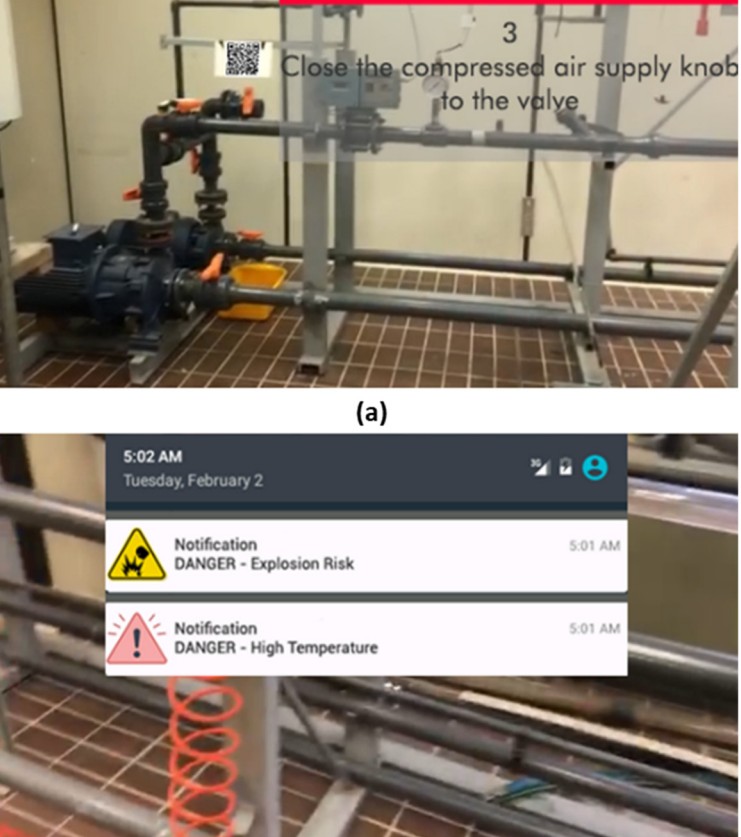

**Figure 5.** Examples of smart glasses function: (**a**) markers; (**b**) emergency notifications.

### 4.3.3. User Interface

The platform interface is probably the most important component of the system, as it ensures that even inexperienced operators can use the system. Through this interface, it is possible to see the sensor readings in real time; evaluate the deviation between the actual behavior of the ejector and the simulation; and, finally, identify anomalies. The interface is equipped with a 3D system model, whereby anomalies are directly reported to facilitate the identification of such issues. The web interface is divided into three columns, as shown in Figure 6. The first column, on the left, includes the 3D visualization of the plant updated in real time and the sensor indicators below it. In the central column are the graphs of the sensors over time. The top column displays the response of the neural network to anomalies. Specifically, the blue line indicates the maximum stability threshold, while the red line indicates the presence of an anomaly. The colored LEDs identify the individual faults next to the graph.

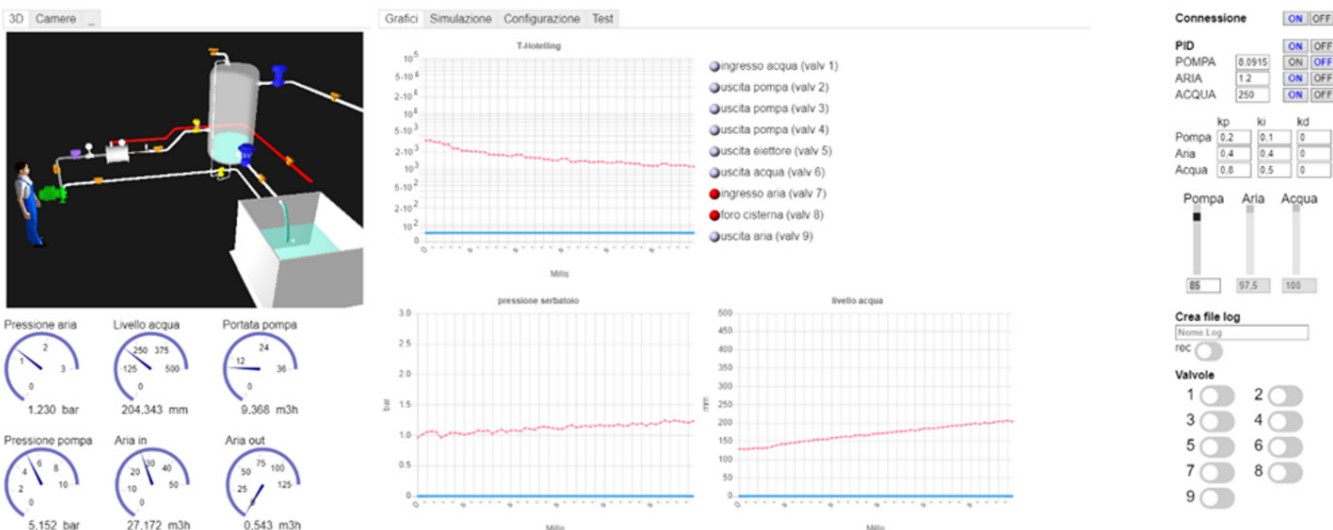

**Figure 6.** The realized user interface: in particular, the blue line indicates the maximum stability threshold, while the red line indicates the presence of an anomaly.

## 5. Course Structure

Based on the explanations in Section 4, once all these preliminary tasks and analyses have been performed, it is possible to define the final course structure, as described below. Lessons can be classified into three types: theoretical (TL), practical (PL), and evaluated practical (EPL). Specifically, practical lessons refer to activities that require the practical application of what has been learned in the theoretical lessons with the active support of a trainer, and the evaluated practical lesson is the final practical activity in which the trainer is present only in a supervisory capacity to judge the execution of a specific task by the participants.

Due to national and internal university regulations related to the COVID-19 pandemic, theoretical lessons were provided remotely, with one per day for the duration indicated in brackets. Conversely, participants were divided into groups for the practical lessons, and the duration indicated in brackets represents the time dedicated to each of these activities. At the end of each week, the participants answered a short questionnaire to evaluate the training provided and their perceived results (see Table A1 in Appendix A).

In order to develop an efficient course, following the identification of needs and objectives as described above, the course structure was based on the following three objectives:

1.  *Learning objective*: participants should be able to manage the plant in normal working conditions through the technologies introduced in the project;
2.  *Conditions objective*: participants should be able to manage the plant under anomalous working conditions through the technologies introduced in the project;

3.  *Behavioral objective*: participants should be able to manage the plant in normal and anomalous working conditions through the technologies introduced in the project.

Specifically, to evaluate learning objectives 1 and 2, participants were made aware of what was happening and what phase of the training activity they were working on. In contrast, concerning the assessment of learning objective 3, the participants did not know what to expect and were left to decide what to do.

### 5.1. Week 1—Understanding The Project

The aim of the lessons delivered during this week was to provide the participants with basic knowledge of the experimental plant being studied and the regulatory framework in terms of safety.

*   *Lesson 1 (1 h duration, TL):* introduction to the project and training course—the project and training course objectives are described to the participants;
*   *Lesson 2 (2 h duration, TL):* Industry 4.0 and its pillars—a brief introduction to Industry 4.0 principles is delivered to the course participants;
*   *Lesson 3 (2 h duration, TL):* safety in industrial plants—the Italian safety regulations for the plant under study are explained to the participants;
*   *Lesson 4 (2 h duration, TL):* explanation of reference pressure system operation—the experimental plant is described to the participants in terms of its functions and final product;
*   *Lesson 5 (1 h duration, PL):* guided use of the reference system—the experimental plant is put into operation, and the participants carry out basic tasks under the supervision of experienced personnel to better understand the technology.

### 5.2. Week 2—Smart Sensor and Digital Twin: The Plant Core

The aim of this week's lectures was to provide participants with basic knowledge of the primary Industry 4.0 technologies that form the foundation on which the whole project is built. The practical lessons were aimed at further consolidating the acquired theoretical notions.

*   *Lesson 1 (2 h duration, TL):* the IoT world and smart sensors—the landscape in which IoT is a powerful and dominant driver is introduced. The participant is introduced to an increasingly network-oriented world, where smart sensors form the backbone of the interface between users and the multitude of devices that surround us, such as smartphones, wearables, robots, and drones;
*   *Lesson 2 (2 h duration, TL):* digital twins—digital twins are introduced as a key aspect of digital transformation, and it is explained how the accurate virtual replication of physical objects, assets, and systems can increase productivity, streamline operations, and boost profits;
*   *Lesson 3 (1 h duration, PL):* guided use of smart sensors—participants have the opportunity to evaluate how the collection of real-time data on the operation of a "product" is improved and how the integration of the various systems or processes contributes to a better overview, which is useful for monitoring and tracking components and the system;
*   *Lesson 4 (1 h duration, PL):* guided use of the digital twin in the reference plant—participants have the opportunity to compare and evaluate different operation scenarios of the experimental plant, exploring how a DT provides efficient support for decision making thanks to the availability of data and analysis.

### 5.3. Week 3—Maintenance and Plant Safety

The lessons delivered this week aimed to provide participants with basic knowledge of plant maintenance and the digital technologies that can facilitate its implementation. The practical lessons were aimed at further consolidating the acquired theoretical notions.

- *Lesson 1 (2 h duration, TL):* maintenance and predictive maintenance—the concept of maintenance is explored in depth, focusing on the similarities and differences between preventive and predictive maintenance and how they both aim to keep a company's production facilities in perfect working order by anticipating breakdowns. The two maintenance strategies are carried out on different operational levels, although they share the same ultimate goal of avoiding malfunctions and sudden plant breakdowns;
- *Lesson 2 (2 h duration, TL):* augmented reality—participants learn what augmented reality is and why it has opened up a new era in remote maintenance and service processes;
- *Lesson 3 (1 h duration, PL):* guided use of predictive maintenance tools for the reference plant—through guided exercises, participants gain an understanding of how augmented/mixed reality viewers allow operators in the field to share their point of view with operation centers, which in turn can guide them via precise instructions, allowing them to operate hands-free from obsolete paper manuals.

*5.4. Week 4—Practice*

The aim of the lessons delivered during this week, which were mainly practical, was to enable participants to act independently (supervised by expert staff) in the management and control of the system, referring to the theoretical and practical knowledge acquired during the previous weeks.

- *Lesson 1 (2 h duration, TL):* IT tools for plant control and management—participants are introduced to the world of user interfaces to easily manage the most complex technologies for the instrumentation, control, and regulation of machines/plants. Participants explore how these applications encompass the vital functions of the machine, allowing the end-user to increase performance and efficiency through portable devices, logs, usage control, and activity management;
- *Lesson 2 (1 h duration, PL):* guided use of the control and management system for the reference plant—participants are trained in the use of the system and the digital technologies developed for its management and control;
- *Lesson 3 (1 h duration, EPL):* supervised use of the reference plant—participants are divided into groups, which are each tasked with an objective to achieve in the simulation: the creation and detection of anomalies, maintenance, etc. Experienced personnel supervise all activities to ensure that the conditions are safe and that all necessary operations are carried out correctly.

## 6. Results

The course was attended by eight operators in the industrial sector, each with several years of experience. In the following discussion, operators with more than ten years of experience were classified as "experienced", while those who had less than three years of experience or were in an apprenticeship period were classified as "inexperienced". The classification "expert/inexpert" is not intended to be a judgement of the practitioner but is only a way to easily identify the subjects of the analysis. Moreover, based on Prensky's argument [23], operators were classified as digital natives (born after 1980) or digital immigrants (born before 1980). As evidenced in Table 1 it is evident that all the inexpert operators were also digital natives, and all the experts were digital immigrants, so two distinct classes were represented:

1. **Inexepert Native (IN)**—operators with little work experience but extensive knowledge of digital technologies;
2. **Expert Immigrant (EI)**—operators with a lot of work experience but little knowledge of digital technologies.

**Table 1.** Operators' backgrounds.

| ID | Operator Class | Digital Native/Immigrant | Apprenticeship | Years after Graduation | Work Experience (Years) | Pressure Plant Experience (%) | Maintenance Experience (%) | Safety Experience (%) |
|----|---------------|--------------------------|----------------|------------------------|-------------------------|-------------------------------|----------------------------|------------------------|
| Op 1 | Inexpert | Native | 0 | 3 | 3 | 67% | 67% | 33% |
| Op 2 | Inexpert | Native | 0 | 4 | 2 | 50% | 0% | 100% |
| Op 3 | Inexpert | Native | 1 | 2 | 1 | 0% | 100% | 0% |
| Op 4 | Inexpert | Native | 1 | 2 | 1 | 100% | 100% | 0% |
| Op 5 | Expert | Immigrant | 0 | 8 | 10 | 70% | 50% | 50% |
| Op 6 | Expert | Immigrant | 0 | 14 | 12 | 83% | 67% | 33% |
| Op 7 | Expert | Immigrant | 0 | 19 | 18 | 83% | 67% | 33% |
| Op 8 | Expert | Immigrant | 0 | 25 | 22 | 100% | 45% | 41% |

Thus, to make the best use of the characteristics of each class, the groups for the practical activities were composed of an inexpert native and an expert immigrant. At this point, the study aimed to implement a training course for industrial operators in the field of new technologies so that they could continue their work effectively. In addition, a further objective derived from analyzing the questionnaires on the perceived quality of the lessons attended (with self-assessment evaluation questionnaires completed both before and after the activities, as proposed in [3,4]) and the external evaluations of the final activity was to identify a standardized form of the course that is able to meet the needs of all types of participant.

Figure 7 shows the operators' backgrounds concerning the topics covered in the course. Evidently, the Industry 4.0 paradigm was fairly well-known to all participants, since their personal evaluations ranged between medium-low (ML) and medium-high (MH). The overall level of knowledge was supported by the fact that the operators showed varying knowledge depending on the topic, according to the more specific questions.

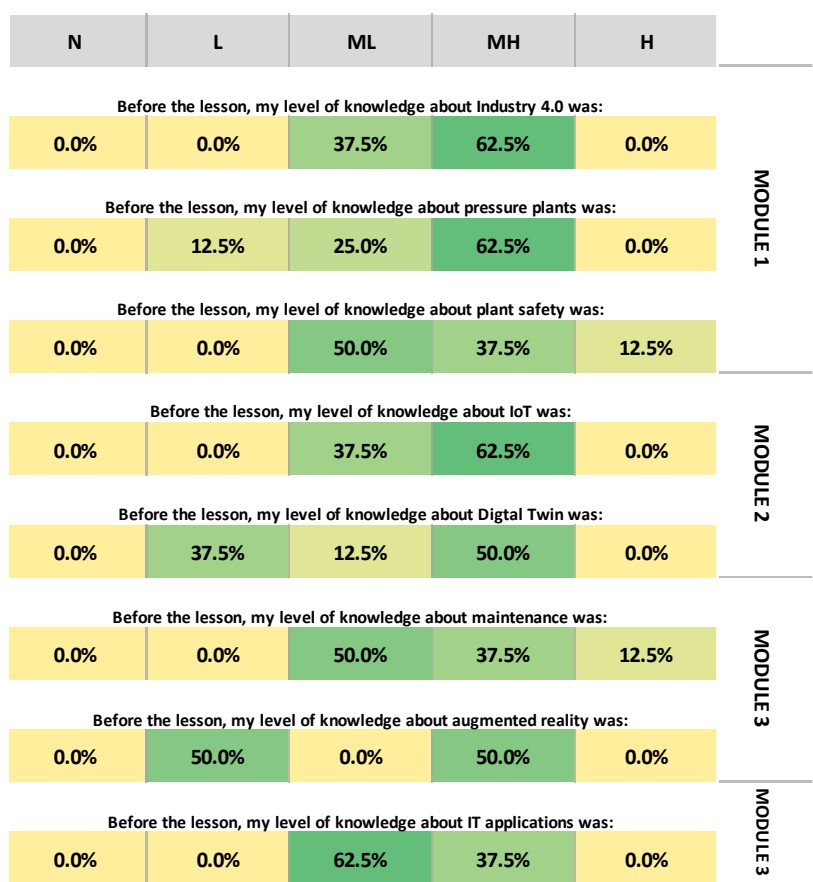

**Figure 7.** General answers concerning the participants' backgrounds. Different colors characterize the different ranges of answers: the closer to 100%, the greener the shade.

The concept of IoT was familiar to all the participants, with their answers ranging between medium-low and medium-high, which was the same for the plant safety and plant IT applications concepts. The participants indicated more fragmented knowledge of the concept of digital twins. Of the participants, 50% claimed to have medium-high knowledge, 12.5% medium-low knowledge, and 37.5% rated their level of knowledge about DT as low but at least claimed to have heard of it. The opposite was the case regarding the concept of plant maintenance, with values varying between medium-low and high, demonstrating the participants' particular attention to this subject, which is essential in the industrial context. A special situation was observed regarding augmented reality. The answers showed that 50% of the participants had only heard about this concept (ML), while the remaining 50% had some practical experience with it.

The variability of the answers confirmed the need to evaluate the answers to each question in relation to the operator's class (EI/IN). As indicated in Figure 8, all the IN operators had some practical experience with augmented reality (100% MH) as a topic of study in their academic careers. Conversely, the EI operators knew a lot about it but had no practical experience (100% L). In the same way, regarding plant safety and maintenance, the EI operators knew that they had in-depth knowledge about the topics, since they resolved such issues daily (75% MH and 25% H). The opposite was the case for IN operators, who understood the theoretical aspects but were just starting to tackle the practical ones.

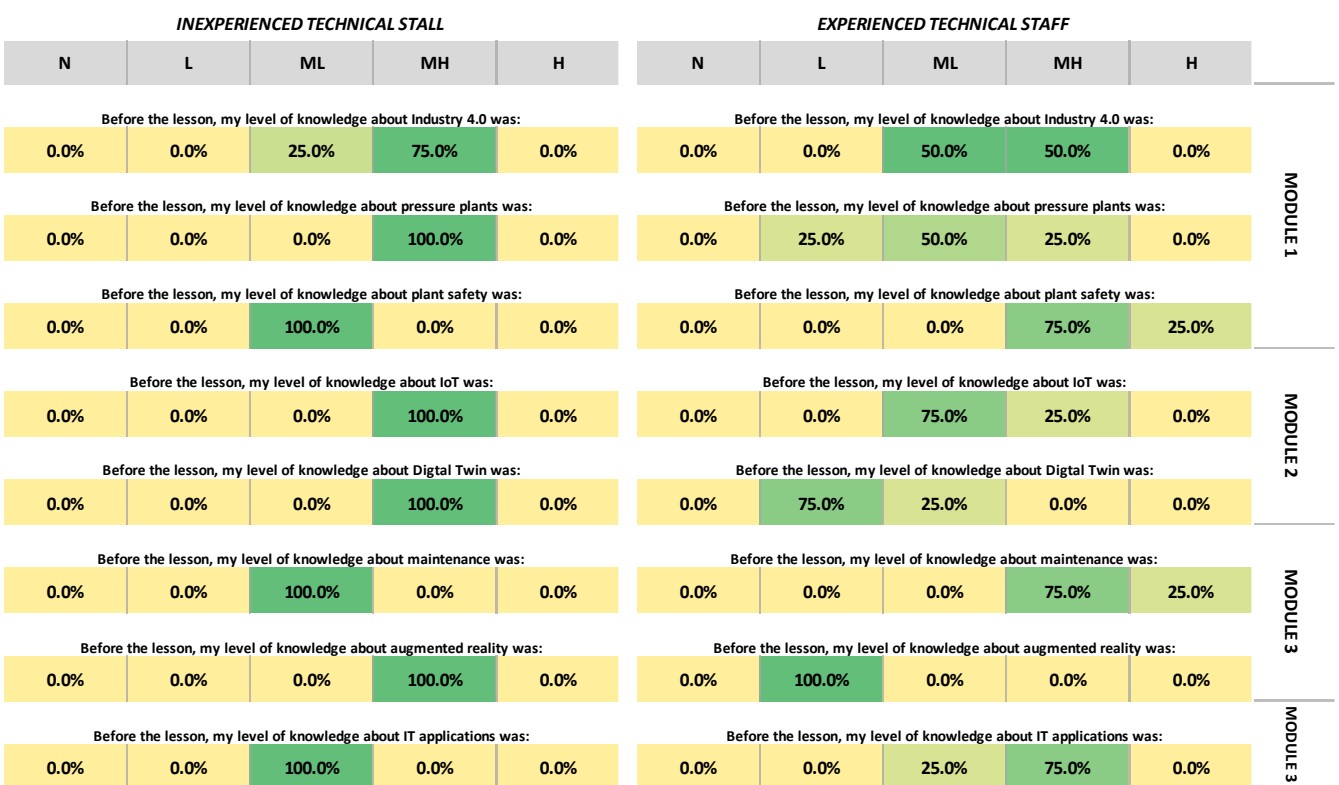

**Figure 8.** General answers concerning the participants' backgrounds according to the operator class. Different colors characterize the different ranges of answers: the closer to 100%, the greener the shade.

As shown in Figure 9, the mean value of the operators' perceived quality level concerning the theoretical and practical activities was MH, with some ML scores attributed to more specific topics and some MH scores to more general ones. It is important to underline how, in the same cases, the low value for a theoretical activity was compensated for by the corresponding practical activity: this was evident in the second module concerning IoT and digital twins.

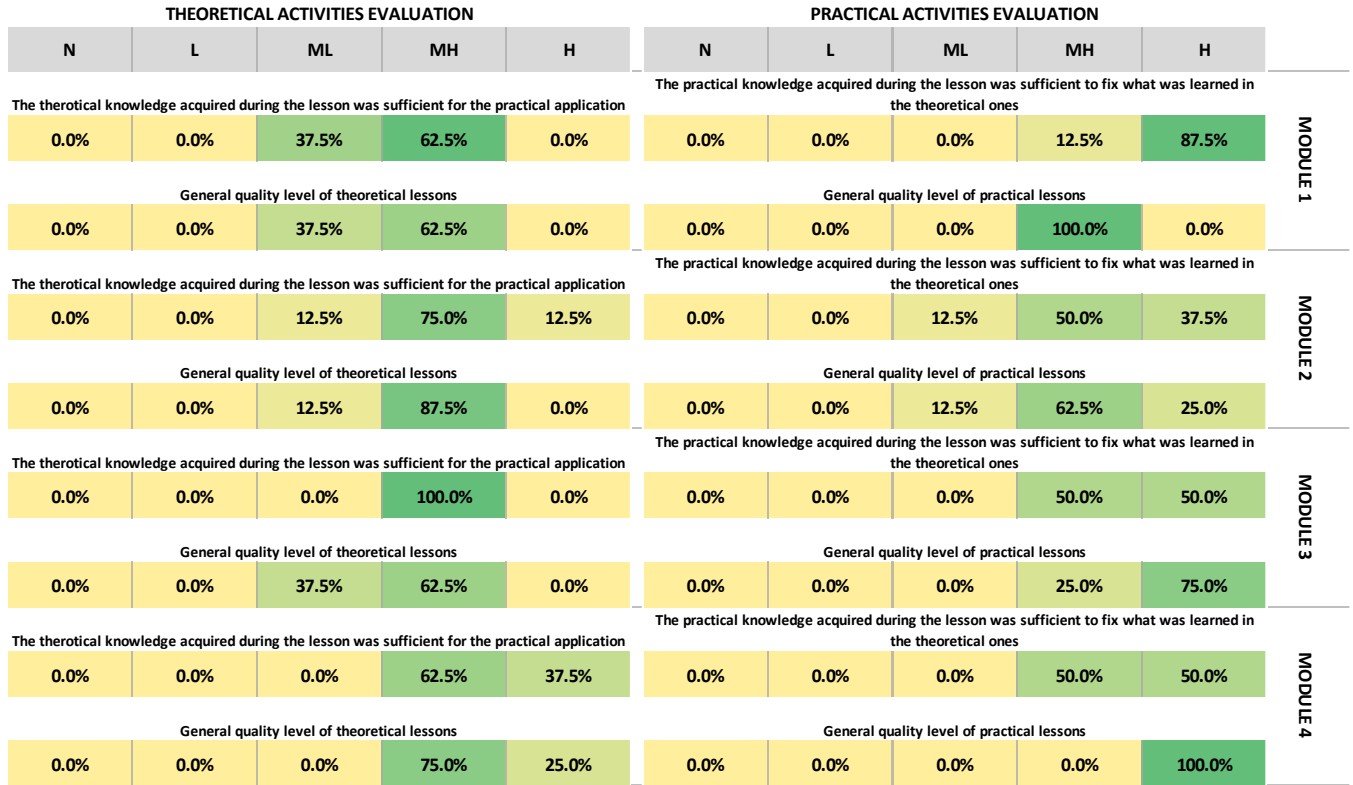

**Figure 9.** Participants' evaluations of the theoretical and practical activities. Different colors characterize the different ranges of answers: the closer to 100%, the greener the shade.

Paying attention to the operators' class, Figure 10 shows that the IN operators scored the quality of the theoretical activities higher due to their young age and the short time that had elapsed since the end of their studies. In contrast, the EI operators, who were older and no longer used to studying, showed greater difficulty understanding the theoretical concepts.

The analysis of Figure 11 further supports this conclusion. Indeed, the EI operators showed greater difficulty in using and applying what they had learned in the theoretical lessons compared to the INs, who gained practical feedback from these tasks, resulting in a higher and more uniform perceived level of quality across all modules.

This experiment was conducted in order to demonstrate and resolve this dichotomy and disseminate the project results. More specifically, as stated in the previous sections, one of the project's objectives was to not only obtain a tool capable of improving the safety conditions of a plant but also to define a corresponding training approach that any operator could understand. The outcome of the evaluation (see Table 2) of the groups showed that the cooperation between an "expert" and an "inexpert" operator resulted in a very high level of group performance. An excerpt of the evaluation scorecard can be found in Appendix A—Table A2.

We compared the supervisors' and the Team 2 operators' evaluations in Tables 2 and 3. The supervisors' average rating for Team 2 was medium-high, highlighting their shortcomings in understanding and applying theoretical concepts, which subsequently led to difficulties in working independently. At the same time, the Team 2 members, particularly the expert, claimed to have had difficulties applying the theory in practice.

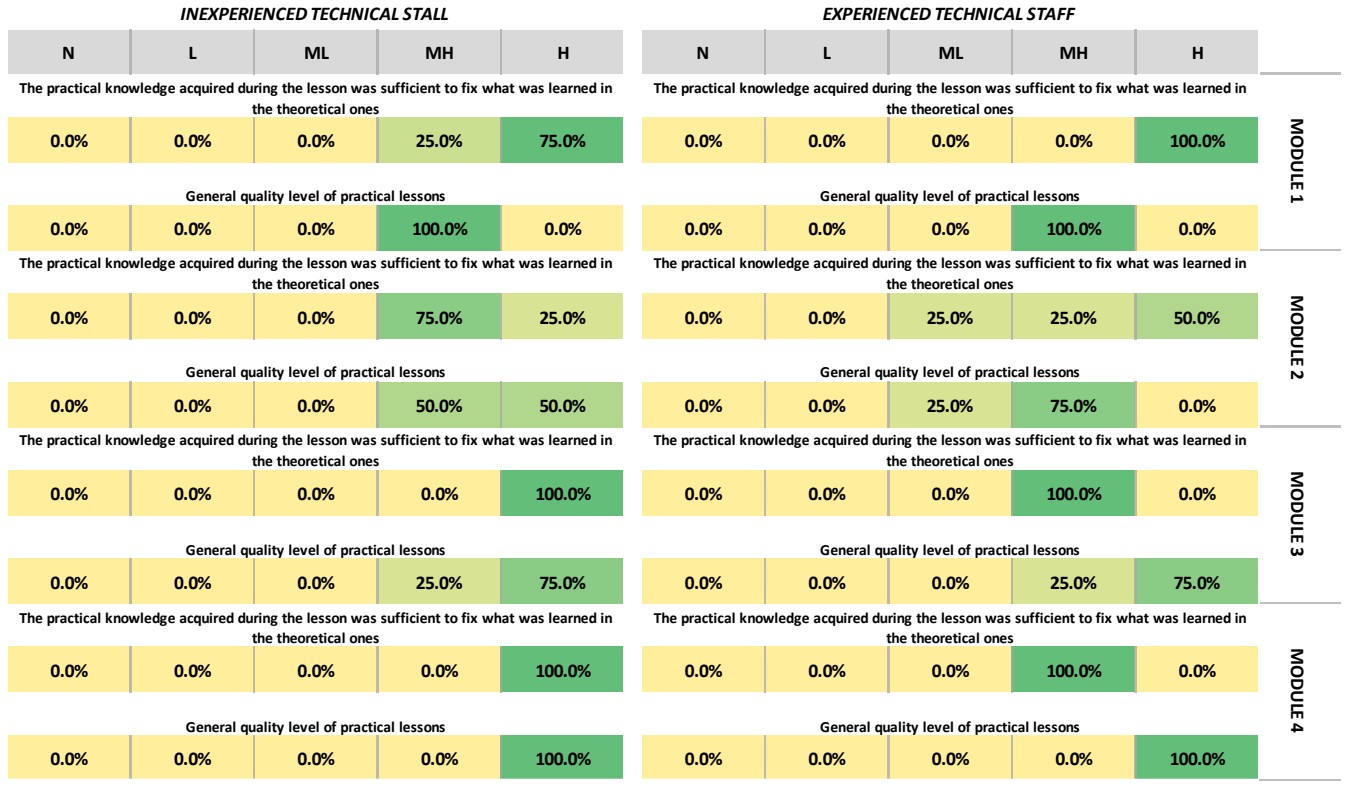

**Figure 10.** Participants' evaluations of the theoretical activities according to operator class. Different colors characterize the different ranges of answers: the closer to 100%, the greener the shade.

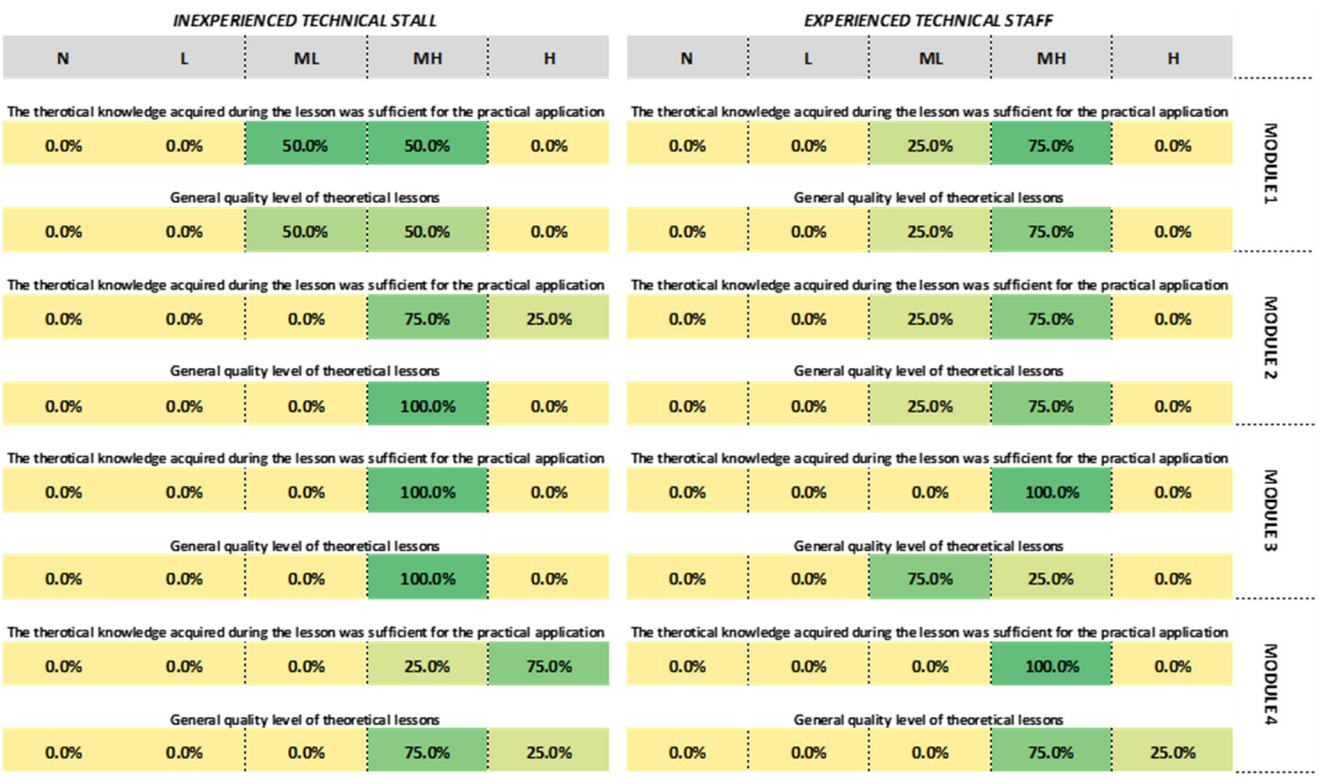

**Figure 11.** Participants' evaluations of the practical activities according to operator class. Different colors characterize the different ranges of answers: the closer to 100%, the greener the shade.

**Table 2.** Teams' evaluations of final practical activity.

| Questions | Team 1 | Team 2 | Team 3 | Team 4 |
|---|---|---|---|---|
| The group demonstrates an understanding of the theoretical concepts covered in the lessons | Medium-high | Medium-low | Medium-low | High |
| The group demonstrates an understanding of the practical concepts covered in the lessons | Medium-high | Medium-high | Medium-high | High |
| The group worked as a team during all practical activities | High | Medium-high | Medium-high | High |
| The group demonstrated autonomy in carrying out the practical tests | Medium-high | Medium-low | Medium-high | Medium-high |
| The objectives required in the practical tests were achieved | Medium-high | Medium-high | Medium-high | High |
| **Final Score** | **Medium-high** | **Medium-high** | **Medium-high** | **High** |

**Table 3.** Team 2 members' evaluation answers.

| Module | Questions | Inexpert | Expert |
|---|---|---|---|
| | Before the lesson, my level of knowledge about Industry 4.0 was: | Medium-high | Medium-low |
| 1 | Before the lesson, my level of knowledge about pressure plants was: | Medium-high | Low |
| | Before the lesson, my level of knowledge about plant safety was: | Medium-low | Medium-high |
| 2 | Before the lesson, my level of knowledge about IoT was: | Medium-high | Medium-low |
| | Before the lesson, my level of knowledge about Digital Twin was: | Medium-high | Low |
| 3 | Before the lesson, my level of knowledge about maintenance was: | Medium-low | Medium-high |
| | Before the lesson, my level of knowledge about augmented reality was: | Medium-high | Low |
| 4 | Before the lesson, my level of knowledge about IT applications was: | Medium-low | Medium-low |

| Module | Questions | Inexpert | Expert |
|---|---|---|---|
| 1 | The theoretical knowledge acquired during the lesson was sufficient for the practical application | Medium-low | Medium-high |
| | General quality level of theoretical lessons | Medium-low | Medium-high |
| 2 | The theoretical knowledge acquired during the lesson was sufficient for the practical application | Medium-high | Medium-low |
| | General quality level of theoretical lessons | Medium-high | Medium-low |
| 3 | The theoretical knowledge acquired during the lesson was sufficient for the practical application | Medium-high | Medium-high |
| | General quality level of theoretical lessons | Medium-high | Medium-low |
| 4 | The theoretical knowledge acquired during the lesson was sufficient for the practical application | High | Medium-high |
| | General quality level of theoretical lessons | Medium-high | Medium-high |

| Module | Questions | Inexpert | Expert |
|---|---|---|---|
| 1 | The practical knowledge acquired during the lesson was sufficient to fix what was learned in the theoretical ones | High | High |
| | General quality level of practical lessons | Medium-high | Medium-high |
| 2 | The practical knowledge acquired during the lesson was sufficient to fix what was learned in the theoretical ones | Medium-high | Medium-low |
| | General quality level of practical lessons | Medium-high | Medium-low |
| 3 | The practical knowledge acquired during the lesson was sufficient to fix what was learned in the theoretical ones | High | Medium-high |
| | General quality level of practical lessons | High | High |
| 4 | The practical knowledge acquired during the lesson was sufficient to fix what was learned in the theoretical ones | High | Medium-high |
| | General quality level of practical lessons | High | High |

This study analyzed the different learning paths of operators, combining their previous experience in the field and the lessons learned from the proposed course. Specifically, theoretical lessons were proposed for all of the students. In contrast, practical lessons were

organized by creating heterogeneous groups on the basis of the theoretical and practical knowledge of the components. Indeed, an incorrect operation action, such as opening a valve on a system that is supposed to be empty or welding on a container that still holds flammable vapors, can lead to serious risks of injury or even accidents that can affect the whole company. Thus, the digitalization of the proposed experimental plan could be useful for risk reduction. At the end of the course, the participants' performance was assessed through a self-evaluation questionnaire and a performance review from the learning plant supervisor. The results underlined a good general level of knowledge concerning Industry 4.0; though the concepts of digital twins and augmented reality were less widespread, the assessment showed in-depth knowledge in the field of maintenance. From the cross-referencing of the responses, it was also found that operators with more practical experience often sacrificed knowledge of the theoretical aspects and vice versa.

Similarly, applying the concepts learned during the theoretical lessons was easier for the inexperienced operators, who were certainly more accustomed to theoretical study and less influenced by the practical skills learned in the field over years of work. The differences found between these learning paths testify to the importance of applying learning tools that meet the needs of both experienced and inexperienced practitioners. Creating heterogeneous groups in terms of experience represents a key tool for comprehensive training, with a view to improving the safety and reliability of plants by integrating advanced tools from the world of Industry 4.0 and digital twins with broader practical applications.

## 7. Discussion

The application of Industry 4.0 and digital twins, although more evident in the manufacturing industry, is gaining a foothold in the process industry, generating benefits that justify past investments and motivate future ones. For these reasons, the oil and gas industry is aiming to become increasingly innovative and implement smart technologies to increase the levels of operational efficiency and resource utilization, while minimizing health, safety, and environmental risks and, last but not least, operating costs. However, as previously mentioned, although innovation can be seen as an opportunity for companies, it may become a problem if operators are not properly trained in the use of new technologies. Hence, it is important to enhance operators' competencies and capabilities in these areas through specific training courses oriented towards the practical learning of these new technologies, so that the reliability and safety of process plants can be strengthened and the operating conditions can improve.

This was precisely the aim of the proposed article: to discuss the results obtained in the design and implementation of a training course to educate operators in the oil and gas sector in the use of this technology. Young and old employees were involved in the study to elucidate the differences between these two operator classes regarding the acquisition of new skills. The identified differences testify to the importance of applying learning tools that are able to meet the needs of both experienced and inexperienced practitioners.

Since education's goal is to provide the finest information transfer service in a manner that is in line with business environment advancements, the proposed training course aims to create a link between research, technological, and organizational aspects. Operator participation is necessary for all operations, and therefore the better equipped the operators are to meet business needs, the easier it will be to incorporate them into business processes.

In fact, this study aimed to fill operators' knowledge gap regarding the use of new and cutting-edge technologies that are becoming increasingly present in the oil and gas sector. Doing so will contribute to the full achievement of the company's objectives and, at the same time, the safety conditions of both the plant and its operators.

This study had several limitations. Firstly, the COVID-19 pandemic limited the initial implementation of this course to a small number of participants (eight in total), due to the university's internal rules. Indeed, such a small number of participants did not allow for statistically valid results, so we were only able to provide qualitative observations concerning the appreciation of the course through the evaluation questionnaires analysis.

In addition, the practical and theoretical lessons were split between face-to-face and distance learning, resulting in a varying level of attention among participants, especially those no longer used to study. If training course participants, either 'students' or 'teachers', lack motivation and inner strength, the course itself has no basis and will not lead to any results. Motivation is the most important aspect of training, and all participants must realize that they are not 'players for fun' but that they can benefit from the knowledge gained across the whole process. In particular, inexpert operators (young employees) attributed a higher level of quality to the theoretical activities due to their young age and the short time that had elapsed since the end of their studies. In contrast, expert operators (senior operators), who were older and no longer used to studying, showed greater difficulty in understanding the theoretical concepts.

The end of the COVID-19 emergency will make it possible to organize more events, involving a greater number of operators in the oil and gas sector and allowing the evaluation of similarities and differences between the groups including statistical significance through structured procedures (e.g., ANOVA tests). Moreover, conducting both practical and theoretical lessons in person will favor constant involvement and interest from all the operators throughout the activities. Furthermore, highlighting the discrepancies between each group in terms of attention and not just knowledge of the technology will allow trainers to develop activities that maintain a high level of concentration and interest. Doing so will make it possible to overcome the abovementioned limitations.

**Author Contributions:** Conceptualization, G.M. (Giovanni Mazzuto), S.A., G.M. (Giulio Marcucci), F.E.C. and M.B.; investigation, G.M. (Giovanni Mazzuto), S.A., G.M. (Giulio Marcucci) and F.E.C.; methodology, G.M. (Giovanni Mazzuto), S.A., G.M. (Giulio Marcucci) and F.E.C.; project administration, M.B.; resources, G.M. (Giovanni Mazzuto), S.A., G.M. (Giulio Marcucci), F.E.C. and M.B.; supervision, M.B.; validation, F.E.C. and M.B.; visualization, G.M. (Giovanni Mazzuto), S.A. and G.M. (Giulio Marcucci); writing—original draft, G.M. (Giovanni Mazzuto), S.A. and G.M. (Giulio Marcucci); writing—review and editing, F.E.C. and M.B. All authors have read and agreed to the published version of the manuscript.

**Funding:** This research received no external funding.

**Informed Consent Statement:** Informed consent was obtained from all subjects involved in the study.

**Data Availability Statement:** Not applicable.

**Acknowledgments:** The authors wish to thank the participants of the study for their willingness to contribute to this research. The authors greatly appreciate their willingness to meet for the extended interviews and to share their thoughts about their experience.

**Conflicts of Interest:** The authors declare no conflict of interest.

## Appendix A

**Table A1.** End-of-module evaluation questionnaire for participants.

| Questions | | | | | |
|---|---|---|---|---|---|
| Before the lesson, my level of knowledge about the topics covered was: | N | L | ML | MH | H |
| The theoretical knowledge acquired during the lesson was sufficient for the practical application | N | L | ML | MH | H |
| The practical knowledge acquired during the lesson was sufficient to fix what was learned in the theoretical ones | N | L | ML | MH | H |
| General quality level of theoretical lessons | N | L | ML | MH | H |
| General quality level of practical lessons | N | L | ML | MH | H |

**Table A2.** End-of-module questionnaire for evaluators.

| Questions | | | | | |
|---|---|---|---|---|---|
| The group demonstrates an understanding of the theoretical concepts covered in the lesson | N | L | ML | MH | H |
| The group demonstrates an understanding of the practical concepts covered in the lesson | N | L | ML | MH | H |
| The group worked as a team during all practical activities | N | L | ML | MH | H |
| The group demonstrated autonomy in carrying out the practical tests | N | L | ML | MH | H |
| The objectives required in the practical tests were achieved | N | L | ML | MH | H |

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
