# Peer review of "Learning-by-Doing Safety and Maintenance Practices: A Pilot Course"

_sustainability, doi:10.3390/su14159635_

Round 1

Reviewer 1 Report

The article deals with an interesting and relevant topic. There are places to fix:

1. It is necessary to emphasize in the introduction what is the gap between what has already been done from a scientific point of view and what you are doing. It is also recommended

 highlight the main subject of the article.

2. In the literature review, it is also recommended to examine the following articles:

Čižiūnienė Kristina; Batarlienė Nijolė. Research on the improvement of industrial practices of transport and logistics students: case study in Lithuania. Transport. Vilnius: VGTU Press. ISSN 1648-4142. vol. 34, iss. 5 (2019), p. 539-547

Bazaras Darius; ÄŒižiÅ«nienÄ— Kristina; Palšaitis RamÅ«nas; Kabashkin Igor. Competence and capacity-building requirements in transport and logistics market. Transport and telecommunication journal. Warsaw: De Gruyter Open Ltd. ISSN 1407-6160. Vol. 17, no. 1 (2016), p. 1-8.

3. Correct the titles of Chapter 3 and Subsection 3.3.2 (make them more scientific).

4. The name of the 2nd picture is written with a lowercase letter.

5. Subsections 4.1-4.4 are too small in scope and should be logically combined into one section.

6. The methodology of the article must be clearly distinguished and named, i.e. there must be a section - Methods and methodology.

7. It is necessary to comment on Figure 3 whether "smart glasses" are regulated (adjusted) taking into account a person's personal physiological characteristics (e.g. myopia, farsightedness, etc.).

Reviewer 2 Report

The Authors deal with an interesting topic and the present paper describes very important issue. My main issue is that the structure of the paper is very poor and does not represent a scientific research paper. Unfortunately, there are several insufficiencies that need to be improved. 

1. the title is interesting, although it may be a bit ambitious and too long.

2. the paper is not correct prepared. Authors did not use template. Lack of some parts, especially ad the end of the paper: Supplementary Materials; Author Contributions; Funding; Institutional Review Board Statement; Informed Consent Statement; Data Availability Statement; Acknowledgments and Conflicts of Interest.

3. The introduction (Section 1) describe the research background very poor. The articles cited are not known in the international literature, they have no citations. Literature review is a separate section (section 2).

4. The manuscript is over structured with no real stated connection between the sections. The authors are urged to rearrange their sections and use the IMRAD structure.

5. I do not see description of used methodology/methods. Please add new section: Methodology. This section should be dedicated to describing this methodology and what you did in your paper. The methodology should be described and be solid enough such that any other person using the same procedure will could repeat the research. 

6. in this paper I do not see very important section: Discussion. Unfortunately, this paper does not point out the shortcomings of past research to show the value of this research. Add a strengths and weaknesses section and limitations section of this research to the new section: Discussion. The discussion should refer to other studies, indicate the shortcomings of the research. The research has some limitations. The manuscript should highlight some of these limitations.

7. in the conclusion, the manuscript should discuss the practical applications and implications of the research.

8. possible areas for future research should be discussed.

9. I believe that this is not a scientific article.

Overall, at the moment the manuscript does not reach the desired level for publishing. I strongly urge the Authors to reconsider the above-mentioned comments, rewrite the paper accordingly, and resubmit

Reviewer 3 Report

The author presents an educational approach for learning Industry 4.0 concepts to mainte- 8 nance and safety operators involved in process industry.

1.That's a good job, but I think the summary should be reorganized. For example, why to do it, how to do it, and what is the significance.

2.The structure of the paper needs to be adjusted, such as establishing a model first, and then analyzing the case.

3.The conclusion needs to be refined. At present, this writing method is lengthy and not readable.

Round 2

Reviewer 1 Report

Worked well. However, I want to draw your attention to the fact that in the 7th source, 2 surnames are spelled incorrectly:

CiziunienÄ— must be ÄŒižiÅ«nienÄ— and Palsaitis must be Palšaitis

Author Response

The spelling errors have been corrected.

The authors want to thank the reviewer for all the precious suggestions.

Reviewer 2 Report

Thank you to the authors for improving the paper.

All comments have been taken into account in the submitted version of the article.

Author Response

The authors want to thank the reviewer for all the precious suggestions.